# Identification and Validation of eRNA as a Prognostic Indicator for Cervical Cancer

**DOI:** 10.3390/biology13040227

**Published:** 2024-03-29

**Authors:** Lijing Huang, Jingkai Zhang, Zhou Songyang, Yuanyan Xiong

**Affiliations:** 1MOE Key Laboratory of Gene Function and Regulation, School of Life Sciences, Sun Yat-sen University, Guangzhou 510275, China; huanglj65@mail2.sysu.edu.cn (L.H.); zhangjk9@mail2.sysu.edu.cn (J.Z.); 2Key Laboratory of Ophthalmology, Zhongshan Ophthalmic Center, Sun Yat-sen University, Guangzhou 510060, China

**Keywords:** eRNA, CESC, prognosis

## Abstract

**Simple Summary:**

Cervical cancer is the fourth most common cancer among women worldwide. The current survival rate of patients with recurrence and metastasis is less than 16.8%. Our research will establish a prognostic model regarding eRNA, which can effectively classify patient prognostic risk. It provides clues for the study of molecular mechanisms in high-risk patients and also provides some potential drugs for the personalized treatment of patients.

**Abstract:**

The survival of CESC patients is closely related to the expression of enhancer RNA (eRNA). In this work, we downloaded eRNA expression, clinical, and gene expression data from the TCeA and TCGA portals. A total of 7936 differentially expressed eRNAs were discovered by limma analysis, and the relationship between these eRNAs and survival was analyzed by univariate Cox hazard analysis, LASSO regression, and multivariate Cox hazard analysis to obtain an 8-eRNA model. Risk score heat maps, KM curves, ROC analysis, robustness analysis, and nomograms further indicate that this 8-eRNA model is a novel indicator with high prognostic performance independent of clinicopathological classification. The model divided patients into high-risk and low-risk groups, compared pathway diversity between the two groups through GSEA analysis, and provided potential therapeutic agents for high-risk patients.

## 1. Introduction

Although cervical cancer screening and preventive human papillomavirus (HPV) vaccination are available in many countries, cervical squamous cell carcinoma and cervical adenocarcinoma (CESC) remain among the leading causes of death among women worldwide [1,2]. In recent years, HPV vaccination has been promoted to prevent cervical cancer, and the number of cervical cancer patients has declined from the 2023 American Cancer Society statistics. At the same time, early detection of cervical cancer through screening allows patients to receive timely treatment, greatly improving their long-term survival rate. However, patients with recurrence or metastasis of cervical cancer still face a serious situation, with a 5-year survival rate of less than 16.8% [3].

Currently, tumor–node–metastasis (TNM) staging is used to determine the size and spread of tumors and is the main indicator of CESC prognosis [3]. However, TNM staging is based solely on anatomical methods, such as physical diagnosis, imaging, laboratory tests, and pathological examinations, and does not reflect tumor heterogeneity [4]. Therefore, in order to individualize treatment for cervical cancer patients properly, it is crucial to identify novel and potent prognostic indications [5].

Enhancer RNA (eRNA) is a class of non-coding RNA transcribed by enhancer regions and is a key regulatory element of gene expression [6,7]. Numerous eRNAs have been identified in human cells, and their contribution to the development of various forms of cancer is known. For instance, eRNA has been found to affect cell cycle arrest in several types of cancer [8]. Recent studies on the pathological mechanisms of eRNAs and their potential applications in human diseases, such as cancer [9], neurodegenerative diseases [10,11], cardiovascular diseases, and metabolic diseases, have helped us understand how eRNAs are involved in disease processes and better comprehend their role in diagnosis, prognosis, or treatment [12].

Studies have shown that eRNA can serve as a prognostic indicator for cancers such as head and neck squamous cell carcinoma [13] and hepatocellular carcinoma [14]. The prognosis and treatment role of eRNA in CESC is of great interest. Given that some studies have described the expression profile and function of eRNA [15,16,17,18], this provides a basis for studying the relationship between eRNA and CESC and screening suitable eRNAs as prognostic markers or drug treatment targets for CESC.

To address the aforementioned problems, we conducted bioinformatics analysis to analyze eRNA. We identified eight eRNAs as a prognostic marker set, as eRNA plays a crucial role in tumors. This marker set is independent of clinical indicators and can effectively distinguish between two subgroups of patients with different prognoses in both the test and training sets. We also examined the differences between the two subgroups, which the 8-eRNA marker set separates, in terms of gene enrichment and pharmacological therapy. We believe that this powerful prognostic marker set will improve risk classification and provide more effective and accurate therapy for cervical cancer patients.

## 2. Materials and Methods

### 2.1. Data Collection and Preprocessing

We obtained the gene expression data and patient clinical information for CESC from the TCGA data portal. The eRNA expression matrix was obtained from the TCeA [19] portal, which integrates eRNA expression data from TCGA, GTEX, and CCLE samples. This collection includes 11 normal samples and 307 tumor samples from CESC. We eliminated samples that lacked all available survival data, resulting 304 CESC patients being included for further research (Appendix A). The clinical follow-up information for CESC patients downloaded from TCGA includes medication, age, survival status, and more. For samples from patients that have died, “days to death” represents the survival time. For samples from patients that have not died, “days to last follow-up” represents the survival time. We used the R package “caret” to randomly divide the 304 samples into a training cohort of 213 samples and a test cohort of 91 samples, while ensuring similar proportions of survival and death between groups (Table 1). The prognostic model is built using the training set, and the reliability of the model is tested using the validation set.

### 2.2. Differential Expression Analysis of eRNAs

For all samples, a total of 302,951 eRNAs were initially considered. After filtering out low-expression data, 55,384 eRNAs were retained for further analysis. Differential expression analysis of tumor and normal eRNAs using the “limma” package in R yielded 7936 differential eRNAs with |log2FC| > 2 and FDR < 0.001 as thresholds (Appendix A). Specifically, the low-expression data were filtered and a log2-fold change was calculated for each eRNA. The analysis then utilized the core steps of limma, including voom, fit, eBays, and other linear regression and difference calculation steps. The output results were sorted based on *p*-value to determine the significance of the differential expression for each eRNA. The Sangerbox analysis tool [20] was used for this step. These calculations were performed using the eRNA expression spectrum dataset obtained in our study.

### 2.3. Establishing and Evaluating Prognostic eRNA Models

To begin, an initial analysis was conducted using univariate Cox hazard analysis to evaluate 7936 differential eRNAs. The LASSO approach was then employed to refine the eRNAs. Subsequently, the remaining variables were filtered using multivariate Cox hazard analysis, resulting in an 8-eRNA model. Risk evaluations for each individual were determined by multiplying the regression coefficients of the 8 eRNAs with their expression levels. The median risk score was used as the threshold to divide the total sample into two subgroups (Appendix A). KM curves and ROC curves were utilized to assess the performance of the model. Finally, the model was applied to test cohorts and overall cohorts to determine its universality. A two-sided log-rank test with a significance level of *p* < 0.05 was employed.

To further test the robustness of the model, TNM incomplete data were initially removed, resulting in 74 patient samples for the test group, 180 patient samples for the training group, and 254 patient samples for the binding group (Appendix A). We used multivariate Cox regression to evaluate the risk-score and clinical information and found that the *p*-values of risk score are significant in the training set, test set, and overall set, proving that the model is robust. The ROC curve compared the performance of metrics using “survivalROC” package assessment and found that the model outperformed other clinical features. In addition, to facilitate the prediction of 1-, 2-, 3-, 5-, and 10-year overall survival (OS) probabilities in CESC patients, nomograms were developed using the “rms” R package.

### 2.4. Gene Set Enrichment Analysis (GSEA)

The model divided 304 CESC patients into two subgroups, and we conducted a differential expression analysis of genes between the two groups (Appendix A). We screened 141 genes that showed differential expression, using thresholds of |log2FC| ≥ 1.5 and *p* < 0.05 (Appendix A). These differentially expressed genes were then used for GSEA analysis in both groups. GSEA KEGG enrichment analysis was performed using the “Dose” package in R, and the results showed the top six enriched pathways in both groups.

### 2.5. Estimation of Drug Response in Clinical Samples

We used the “oncoPredict” R package to predict drug IC50 values in CESC patients. Gene expression data and drug IC50 data for the drug prediction training set were taken from Genomics of Drug Sensitivity in Cancer (GDSCv2), which by default used 10-fold cross-validation. As a gauge of drug sensitivity, both datasets give the estimated IC50 values (Appendix A). The drug high- and low-expression group data were plotted by prism software, and its significance was tested using the Mann–Whitney test.

### 2.6. Statistic Assessment

R version 4.2.3 was used for all statistical analyses, including the following packages: rjson, limma, caret, Matrix, survival, survminer, ggplot2, ggpubr, rms, survivalROC, Dose, oncoPredict, and others. When the model distinguishes between high- and low-risk groups, we choose the median value as the distinguishing point because the patient risk score is not normally distributed. Statistical significance is defined as a test value of *p* < 0.05 [21].

## 3. Results

### 3.1. A Total of 7936 eRNAs Are Differentially Expressed in CESC

To facilitate further analysis, we divided the CESC patients into a training group and a testing group following a 7:3 ratio (Table 1). We then utilized the “limma” package to perform differential analysis of eRNA expression levels between the normal and tumor samples. As a result, we identified a total of 7936 eRNAs that were differentially expressed. The selection criteria were based on an FDR < 0.001 and |log2FC| > 2 indicators (Figure 1a; Appendix A). Additionally, we generated a visual representation of the expression heatmap for the first 50 eRNAs (Figure 1b).

### 3.2. An 8-eRNA Prognostic Model with Good Performance Was Constructed

In the training set, we analyzed 7936 differentially expressed eRNAs using univariate Cox hazard analysis. Our findings revealed that 36 eRNAs were substantially associated with OS (*p* < 0.001) (Appendix A). Additionally, we filtered out 15 eRNAs using LASSO regression analysis and 10-fold cross-validation (Figure 2a,b). Subsequently, we performed multivariate Cox hazard analysis using these 15 eRNAs. The regression algorithm automatically identified the best combination of eRNAs, resulting in eight potential eRNAs: chr1:172108837, chr2:191785234, chr8:144814327, chr9:139695311, chr11:65074820, chr18:46478071, chr20:30801475, and chr21:45165011 (Figure 2c; Table 2). We developed an 8-eRNA prediction model to assess the impact of these potential eRNAs on CESC patient survival based on their expression levels and coefficients obtained from the multivariate Cox hazard analysis. The formula for calculating the CESC prognostic risk score is as follows: Risk score = (0.08172 × chr1:172108837) + (0.59886 × chr2:191785234) + (0.47507 × chr8:144814327) + (−1.29240 × chr9:139695311) + (1.02553 × chr11:65074820) + (0.24535 × chr18:46478071) + (−2.75351 × chr20:30801475) + (0.18672 × chr21:45165011).

The individuals in the training set were separated into long-term and short-term survival subgroups according to the median of the risk assessment score (Figure 2d; Appendix A). The K-M survival curve was used to compare the survival rates of the high-risk and low-risk groups, and the results revealed that patients in the high-risk group had significantly shorter survival times compared to those in the low-risk group (log-rank test *p* < 0.001) (Figure 2e). Additionally, the ROC curve was employed to assess the sensitivity and specificity of the risk score in predicting survival. The area under the curve (AUC) was measured at 0.857 for the 8-eRNA risk evaluation, indicating its ability to accurately predict survival in patients with CESC (Figure 2f).

### 3.3. The 8-eRNA Model Performs Well in the Validation Set

The validation set consists of the test queue and the entire queue. The entire queue is formed by combining the test queue and the training queue. The results of the validation set indicate that the 8-eRNA model can effectively divide the samples of the test set and the entire set into two subgroups based on long-term survival. There are significant differences between these two subgroups, as shown by the risk score heatmap (Figure 3a,d) and KM plot (Figure 3b,e) (logarithmic rank test, test set *p* = 0.0083, entire set *p* < 0.001). Both subgroups show a decrease in survival time as the risk score increases. Furthermore, the ROC analysis of the validation set demonstrates that the 8-eRNA model performs well in both cohorts, with an AUC greater than 0.7 (Figure 3c,f). In conclusion, the CESC validation set confirms that the 8-eRNA model effectively discriminates between the high- and low-risk groups, suggesting its potential as a reliable prognostic model in other validation sets.

### 3.4. The 8-eRNA Model Is Robust

The study investigated the prognostic impact of the 8-eRNA model using various clinical parameters, including age, pathological tumor stage, TMN stage, PD, and risk score (Figure 4a–f, Appendix A). Pathological tumor stage, TMN stage, and risk score were identified as strong predictive factors for OS in both univariate and multivariate Cox hazard analyses. The multivariate Cox hazard analysis showed hazard ratios of 1.516 (1.213–1.894) (Figure 4a) and 1.437 (1.196–1.727) (Figure 4f) for the 8-eRNA model, indicating its robust predictive value for OS in both the training set and the combined set. These findings suggest that, in addition to tumor stage and TMN stage, the 8-eRNA model can be used as an independent prognostic factor for OS in CESC patients.

### 3.5. Prognostic Nomogram Construction

We created nomograms in the train (Figure 5a), test (Figure 5c), combined (Figure 5e) cohorts to estimate the survival of patients (254 CESC patients with six clinical variables known to be evaluated) based on age, pathological tumor stage, TMN stage, PD, and risk score. Subsequently, we assessed the accuracy of the risk model using ROC curves. The AUC values obtained for risk scores in the three cohorts were 0.9 (Figure 5b), 0.74 (Figure 5d), and 0.84 (Figure 5f), respectively. These values were higher than those of the remaining six clinical variables in the cohorts, indicating that the model performed the best. These findings suggest that the 8-eRNA model is a more reliable predictor of OS in CESC patients than prognostic differentiation based on clinical features.

### 3.6. Six Pathways Enriched in High-Risk Populations

In order to investigate how changes in eRNA expression influence prognosis, it is essential to compare the functional and signaling pathway differences between the high-risk and low-risk subgroups, which are grouped based on the 8-eRNA model [22]. We conducted a differential expression analysis between these two groups (Appendix A) and identified 141 genes that were differentially expressed using the GSEA analysis thresholds of |log2FC| ≥ 1.5 and *p* < 0.05 (Appendix A). The visualization of the data presents the first six pathways in both subgroups [23]. The high-risk group showed enrichment of genes involved in biosynthesis of cell cycle, DNA replication, amino acids, primary immunodeficiency, mismatch repair, and the proteasome (Figure 6a). On the other hand, the low-risk group exhibited enrichment of genes associated with butanoate metabolism, circadian rhythm, drug metabolism-cytochrome P450, phenylalanine metabolism, taurine and hypotaurine metabolism, and tyrosine metabolism (Figure 6b). These findings offer insights for further investigations into the molecular mechanism by which eRNA influences signaling pathways and subsequently influences prognosis.

### 3.7. Twelve Potential Therapeutic Agents in the High-Risk Group

Given that chemotherapy is still used as adjuvant therapy in the clinic, we conducted a search for potential drugs that can increase drug sensitivity in high-risk patients. This increase in sensitivity is due to changes in eRNA expression [24,25]. To evaluate chemotherapeutic sensitivity in the TCGA high- and low-risk sets, we utilized the “oncoPredict” R package. This package allowed us to construct a predictive model using cell line data from the Genomics of Drug Sensitivity in Cancer (GDSC) database. As a measure of drug sensitivity, both datasets provided estimated concentration for 50% maximum inhibitory concentration (IC50) values (Appendix A).

Based on the drug prediction results, we identified drugs in the high-risk group that had a median drug IC50 significantly lower than that in the low-risk group (Mann–Whitney test, *p* < 0.05). We then selected 12 representative drugs (Figure 7a–l) from this group. Among these drugs, Osimertinib_1919, Afatinib_1032, Erlotinib_1168, and AZD3759_1915 target EGFR; Trametinib and PD0325901_1060 are MEK inhibitors; SCH772984_1564, ERK_2440_1713, Ulixertinib_1908, and VX-11e_2096 are ERK1/2 inhibitors; IGF1R_3801_1738 targets IGFR1; and Taselisib_1561 targets PI3K inhibitors. These drugs exhibited low predicted IC50 values in the high-risk populations, suggesting their potential for the treatment of CESC patients.

## 4. Discussion

CESC is the fourth most common cancer in women worldwide. However, in many underdeveloped countries, the lack of immunization and screening programs has made diagnosing the disease more challenging [26]. While current treatments like surgery, radiation, and chemotherapy show promise for patients, around 75% of them still experience disease progression or recurrence [27,28]. By exploring the carcinogenic mechanism of CESC and developing new prognostic models, we may discover fresh ideas for treating patients [29]. With advancements in high-throughput technologies like CAGE-seq and RNA-seq, as well as the establishment of eRNA data portals such as HeRA [16], eRic [15] and TCeA, eRNA has emerged as an effective method for identifying biomarkers that can predict survival. Previous studies have demonstrated the use of eRNA in predicting the prognosis of head and neck squamous cell carcinoma [13] and hepatocellular carcinoma [30]. In this study, we have developed a unique 8-eRNA prognostic marker set for CESC and demonstrated its usefulness as a predictive tool. Our work provides new scientific insights into the molecular processes of CESC.

The high-risk subgroup GSEA in the paper primarily focused on the following pathways: cell cycle, DNA replication, biosynthesis of amino acids, primary immunodeficiency, mismatch repair, and the proteasome (Figure 6a). In the drug sensitivity analysis of the high-risk group, the potential drugs were mainly MEK inhibitors, ERK1/2 inhibitors, EGFR-targeted drugs, IGFR1-targeted drugs, and PI3K-inhibitor-targeted drugs (Figure 7a–l). Further analysis of the GSEA and drug susceptibility analysis results revealed that MEK inhibitors and ERK1/2 inhibitors affect the MAPK/ERK pathway. Additionally, EGFR can activate the PI3K-AKT and MAPK-ERK pathways [31], while IGF1R can activate the PI3K-AKT pathway [32]. Downstream of the PI3K-AKT pathway and MAPK-ERK signaling pathway is the cell cycle pathway [33,34]. Therefore, we discovered a strong correlation between the pathways targeted by drugs and the pathways that enrich genes in high-risk groups. This also provides insights into exploring the pathway differences between the two subgroups. It is possible that eRNA may affect the replication and cell cycle processes of cells by influencing pathways such as MAPK-ERK and PI3K-AKT, ultimately impacting the survival prognosis of CESC.

To further understand the eight eRNAs in our model, we obtained a table of eRNA and gene correspondence from TCeA (Appendix A) and identified the genes associated with these eight eRNAs (Appendix A). In the 8-eRNA model, we found that two protective factors, chr9:139695311 and chr20:30801475, as well as one risk factor, chr11:65074820, were significantly significant. Firstly, the gene PAXX, which corresponds to chr9:139695311, is a component of the non-homologous terminal junction (NHEJ) DNA repair pathway [35,36]. Studies have shown a connection between the expression level of PAXX and cancer development in human patients [37], and PAXX has been identified as an independent predictor of colon cancer [38]. Secondly, chr20:30801475 corresponds to the gene KIF3B, which produces a protein that functions as a heterodimer of kinesin family member 3A during mitosis and meiosis, contributing to chromosomal mobility. KIF3B has been identified as a potential therapeutic target for various cancers, including breast cancer [39], pancreatic cancer [40], cervical cancer [41], and oral squamous cell carcinoma [42]. Thirdly, the gene CDC42BPG, which corresponds to chr11:65074820, encodes CDC42BPG serine/threonine kinase. This kinase controls the phosphorylation of MYPT1 and MLC2 [43], promoting the actin contractility necessary for cell invasion as a downstream effector of CDC42 [44] in cytoskeletal recombination. In conclusion, the genes PAXX and KIF3B are associated with DNA repair and cell division, supporting the GSEA study’s findings that the high-risk subgroup may affect the long-term survival of cervical cancer by influencing DNA replication and the cell cycle. Further exploration of the role of chr9:139695311 and chr20:30801475 in CESC is warranted. Additionally, there is currently limited research on CDC42BPG, and further investigation into the role of chr11:65074820 in CESC is also warranted.

Since the tumor microenvironment (TME) plays a crucial role in carcinogenesis, development, and therapy, we investigated the relationship between immunological features and prognostic models [24]. To evaluate the proportions of immune and stromal cell components of TME across the two risk groups (Appendix A), we utilized the ESTIMATE algorithm [45], which is implemented in the R package “estimate”. The algorithm generated the stromal score, immune score, and estimate score. We then plotted the computation results using boxplots and found no significant difference in TME between the two groups (Appendix A). Therefore, our findings suggest that the TME does not have a significant impact on the difference in prognosis between the high- and low-risk groups.

A somatic mutation study was conducted to explore the role of mutations in the high- and low-risk subgroups. Genetic differences between the two groups were examined using somatic mutation data for CESC patients that were taken from the TCGA database. The study revealed that less than 50% of samples in both groups had mutations (Appendix A). The two most commonly mutated genes in both groups were FLG and HSPG2 (Appendix A). The FLG [46,47,48] gene encodes an intermediate filament-related protein that aggregates in the keratin intermediate filament in the mammalian epidermis. The HSPG2 [49] gene encodes the perlecan protein, a large multidomain proteoglycan that plays a crucial role in maintaining the function of the endothelial barrier. It binds to and forms crosslinks with several extracellular matrix elements and cell surface molecules. To compare the frequency of mutations in the first 20 genes between the two sets of samples, a chi-square test was performed. The results showed no significant difference in mutations between the two groups, except for two genes, CRYBG3 and DSG3, *p* < 0.05. CRYBG3 is an anchoring protein that regulates the subcellular compartmentation of protein kinase A (PKA). PKA, in turn, activates transcription factors and downstream transcription, which may be associated with the biosynthetic pathway of amino acids enriched in high-risk groups. DSG3 is a protein found in intercellular desmosome junctions. It interacts with plaque proteins and intermediate filaments, playing a role in cell–cell adhesion [50,51]. DSG3 is also involved in the apoptotic cleavage of cellular proteins and programmed cell death.

Despite providing new insights into eRNAs in CESC, this work has several limitations. First, the structures of the eRNAs are not definitive because we lack information on long-range chromosome interactions, such as high-throughput chromosome conformation capture (Hi-C) [52]. Second, the sample size used in this study is insufficient, which affects the efficiency of the statistical analysis. Third, since we only have access to a limited amount of eRNA expression information, our study lacks external validation [13]. Finally, despite demonstrating the accuracy and stability of the 8-eRNA model, we recognized some limitations, which raises concerns about its generalization.

In conclusion, the 8-eRNA model is a robust biomarker for predicting prognosis in CESC patients. This study provides valuable insights into new potential biomarkers for predicting prognosis and survival in cervical cancer patients. Additionally, it is expected to offer promising opportunities for improving medication therapy [30].

## 5. Conclusions

This study aimed to explore the role of eRNA in the survival of CESC patients and identify potential treatment options for high-risk patients. We developed a prognostic model consisting of eight eRNAs using single-factor Cox regression analysis, LASSO regression analysis, and multifactor Cox regression analysis. The model demonstrated high accuracy and stability, with a C-index of 0.81, and showed good performance in the validation set. Based on this model, we categorized patients into high-risk and low-risk groups, and the difference between the groups was statistically significant (*p* < 0.001). Further analysis using Kaplan–Meier curves, ROC curves, robustness analysis, and nomograms confirmed that this model is an independent prognostic indicator with superior predictive ability compared to clinicopathological classification alone. By comparing the pathway differences between the high-risk and low-risk groups through GSEA, it was revealed that the high-risk group was primarily enriched in tumor-related pathways, including cell cycle, amino acid biosynthesis, and DNA replication. It was observed that the expression of eRNA may influence these signaling pathways by modulating gene expression, thus playing a crucial role in the development and progression of CESC and representing a potential therapeutic target. Lastly, using the GDSC database, we identified potential therapeutic drugs for high-risk patients, such as afatinib, trametinib, and urotinib. By influencing gene expression, eRNA may enhance the sensitivity of these drugs, thereby improving personalized treatment outcomes and overall survival rates. In conclusion, this study provides insights into the prognostic significance of eRNA in CESC patients, introduces new prognostic markers, and identifies potential therapeutic drugs for high-risk patients.

## Figures and Tables

**Figure 1 biology-13-00227-f001:**
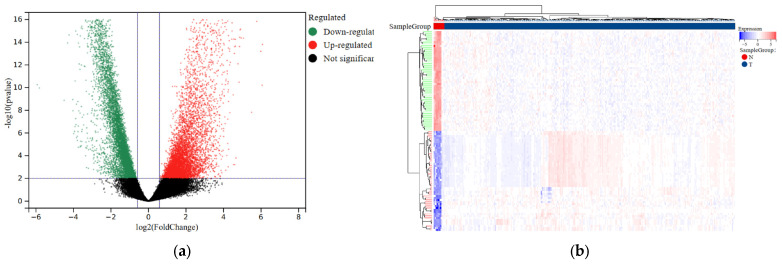
Differential expression analysis of eRNAs in normal and tumor samples of CESC. (**a**) Volcano plot comparing differential eRNA expression analysis in tumor samples and normal samples. (**b**) The heatmap displays the differential expression of the first 50 eRNAs in tumor and normal samples. Upregulation is shown in red, while downregulation is shown in blue.

**Figure 2 biology-13-00227-f002:**
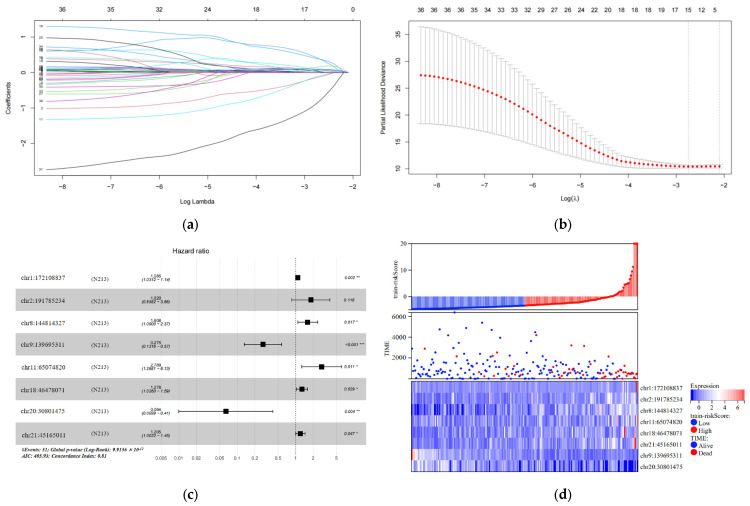
Construction and testing of an 8-eRNA prognostic model for cervical cancer. (**a**,**b**) After performing univariate Cox analysis, the selection of eRNAs was further refined using LASSO regression. (**c**) The optimal combination of 8 eRNAs was determined through multivariable Cox hazard analysis, allowing for the development of a prognostic model. *** *p* < 0.001; ** *p* < 0.01; * *p* < 0.05. (**d**) The training set was divided into two subgroups based on their median risk assessment. The low-risk group is represented in blue (top), consisting of patients who are alive (middle) and have relatively low eRNA expression (bottom). The high-risk group is represented in red (top), consisting of patients who have died (middle) and have relatively high eRNA expression (bottom). (**e**) To distinguish between the high-risk and low-risk groups, the Kaplan-Meier (K-M) survival curve was utilized, along with the median risk score. The log-rank test showed a significant difference (*p* < 0.001). The numbers at risk indicate the number of surviving patients who have not experienced an endpoint event, while the cumulative number of events represents the total number of events at risk. (**f**) The performance of the 8-eRNA models in the training set was assessed using the ROC curve.

**Figure 3 biology-13-00227-f003:**
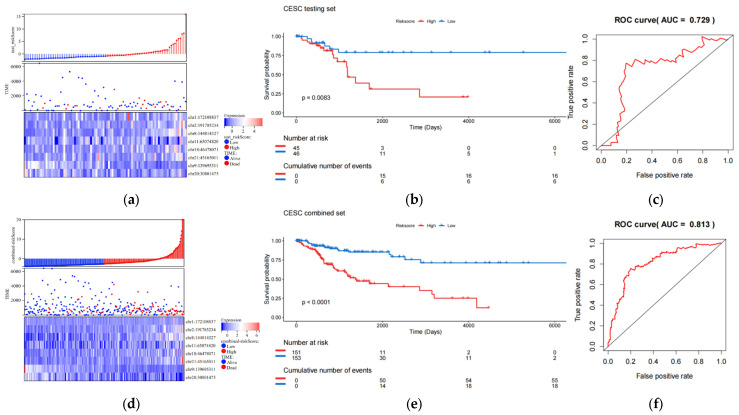
The effect validation of the 8-eRNA model in validation set, the test set, and combined cohorts. (**a**–**c**) The efficacy and performance of the 8-eRNA model were assessed in the test cohort using a risk score heatmap, KM curve, and ROC curve. (**d**–**f**) The same evaluation method was applied to assess the 8-eRNA model in the combined cohort.

**Figure 4 biology-13-00227-f004:**
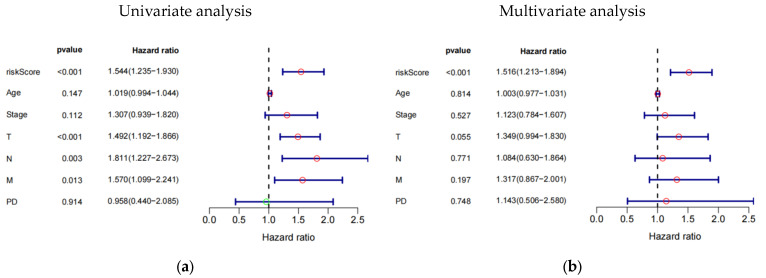
Univariate and multivariate Cox hazard analysis. (**a**,**b**) Cox hazard analysis was performed at both the univariate and multivariate levels to identify clinical characteristics in the training cohort. Green circle: Hazard ratio < 1; Red circle: Hazard ratio > 1. (**c**,**d**) The same methodology was applied to the test cohort. In (**e**,**f**), the analysis encompassed the entire cohort. T: The extent and size of the primary tumor; N: Lymph node dissemination; M: Presence of metastasis; PD: Primary diagnosis.

**Figure 5 biology-13-00227-f005:**
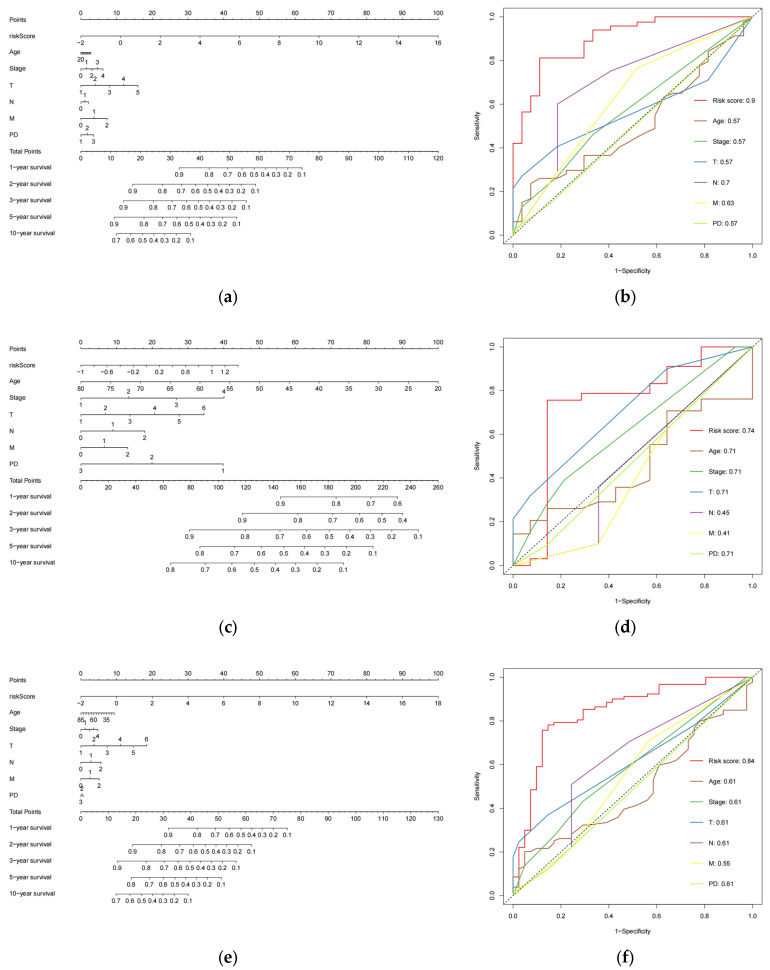
The plot shows the ROC curves for the risk score and clinical variables assessing OS in each cohort, along with the nomograms for predicting OS. (**a**) Nomogram of the training cohort used to predict the OS. (**b**) ROC curves are used to assess the performance of risk scores and clinical variables in predicting OS in the training cohort. Panels (**c**,**d**) demonstrate the same methodology applied to the test set, while (**e**,**f**) showcase the overall set.

**Figure 6 biology-13-00227-f006:**
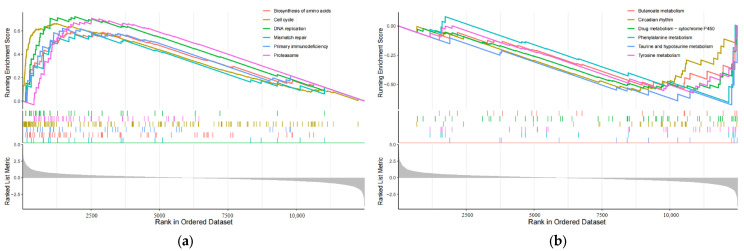
The plot shows the results of GSEA analysis for identifying significantly enriched pathways in the entire cohort. Panel (**a**) represents the analysis performed for the high-risk group, while panel (**b**) shows the results for the low-risk group. Gray area: Signal2noise for each gene.

**Figure 7 biology-13-00227-f007:**
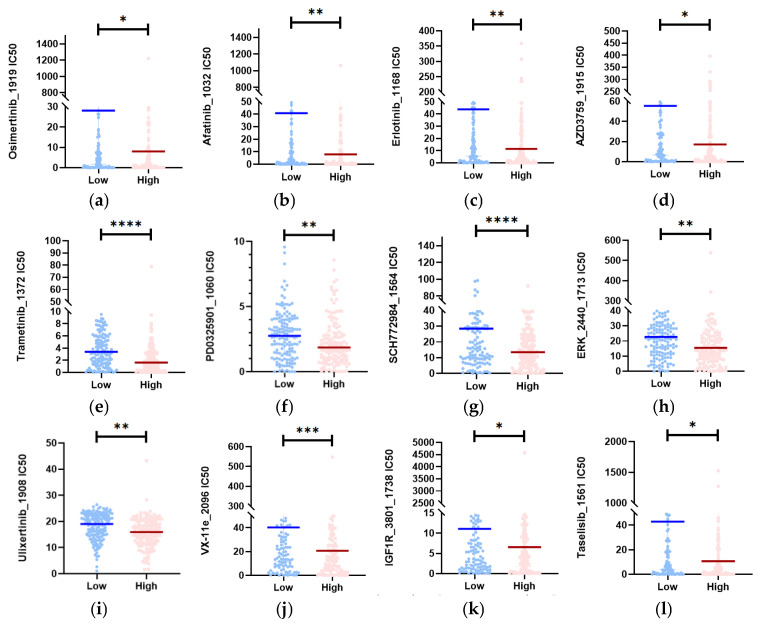
Identification of potential medicines that have higher therapy sensitivity for people in high-risk groups. (**a**–**l**) In the CESC datasets, the estimated IC50 of 12 potential medicines was compared between the two subgroups (Mann–Whitney test). **** *p* < 0.0001; *** *p* < 0.001; ** *p* < 0.01; * *p* < 0.05.

**Table 1 biology-13-00227-t001:** Clinical characteristics of CESC sufferers in each cohort.

Variables		Training Cohort (*n* = 213)	Testing Cohort (*n* = 91)	Combined Cohort (*n* = 304)
Age	Median (Range)	46 (21–80)	47 (20–88)	46 (20–88)
Stage	Stage I	108	55	163
	Stage II	55	14	69
	Stage III	31	14	45
	Stage IV	14	6	20
	N/A	5	2	7
T classification	T1	91	49	140
	T2	58	13	71
	T3	11	9	20
	T4	8	2	10
	TX	16	1	17
	Tis	0	1	1
	N/A	29	16	45
N classification	N0	94	39	133
	N1	39	21	60
	NX	51	15	66
	N/A	29	16	45
M classification	M0	84	32	116
	M1	4	6	10
	MX	92	36	128
	N/A	33	17	50
PD (Primary diagnosis)	Squamous cell carcinoma	176	76	252
	Adenocarcinoma	34	14	48
	Adenosquamous carcinoma	3	1	4

**Table 2 biology-13-00227-t002:** Prognostic eRNA obtained from multivariable Cox regression analysis.

eRNA ^1^	Coefficient	HR ^2^	95% CI ^3^	*p* Value
chr1:172108837	0.08172	1.08515	1.0312–1.14	0.001687
chr2:191785234	0.59886	1.82005	0.8582–3.86	0.118479
chr8:144814327	0.47507	1.60812	1.0900–2.37	0.016644
chr9:139695311	−1.29240	0.27461	0.1316–0.57	0.000571
chr11:65074820	1.02553	2.78858	1.2681–6.13	0.010747
chr18:46478071	0.24535	1.27807	1.0260–1.59	0.028604
chr20:30801475	−2.75351	0.06370	0.0099–0.41	0.003681
chr21:45165011	0.18672	1.20529	1.0022–1.45	0.047293

^1^ enhancer RNA. ^2^ hazard ratio. ^3^ confidence interval.

## Data Availability

The research data used in the article has been provided in the Appendix A.

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
