# Peer review of "Identification and Validation of eRNA as a Prognostic Indicator for Cervical Cancer"

_biology, 2024, doi:10.3390/biology13040227_

Round 1
Reviewer 1 Report
Comments and Suggestions for Authors
In the manuscript "Identification and validation of eRNA as a prognostic indicator for cervical cancer" Huang and co-authors investigated the relationship between eRNA expression and the survival of patients with cervical squamous cell carcinoma and endocervical adenocarcinoma. Data from multiple sources were utilized, including eRNA expression, clinical information, and gene expression data from TCeA and TCGA databases. Through rigorous analysis, 7936 differentially expressed eRNAs were identified using limma analysis. Subsequently, the study conducted univariate Cox hazard analysis, LASSO regression, and multivariate Cox hazard analysis to develop an 8-eRNA prognostic model. This model was validated using various methods demonstrating its high prognostic performance independent of traditional clinicopathological classification. Furthermore, the research categorized patients into high-risk and low-risk groups based on this model and compared the pathway diversity between these groups through GSEA. This analysis revealed potential therapeutic targets for high-risk patients. Overall, the study provides novel insights into the prognostic significance of eRNA in CESC patients, introduces a robust prognostic model, and identifies potential therapeutic options for high-risk patients.
I have some minor questuions:
1) Can you explain the potential mechanisms through which eRNA expression influences signaling pathways and contributes to the development and progression of CESC?
2) What are the limitations of the study, and how might they affect the interpretation and generalization of the findings?
3) Figure 1(b), 2(c) and some others have too small inscriptions and are difficult to read.
4) Correct all typos such as “8-erna” on line 286, etc.
Author Response
请参阅附件。

Reviewer 2 Report
Comments and Suggestions for Authors
The findings, while providing valuable insights, prompt future research directions, exploring additional biomarkers, and potential therapeutic developments in cervical cancer prognosis and treatment.This constitutes a positive point of extreme relevance and must be valued within the scientific scenario, which leads me to congratulate the authors for the topic and research. I am grateful for the opportunity to read this manuscript, which I believe has merit and relevance in the research scenario within cervical cancer. I believe, however, that some may not be addressed:
1. The study explores the potential of an 8-eRNA model as a prognostic marker for uterine squamous cell carcinoma and adenocarcinoma (CESC). Despite demonstrating the accuracy and stability of the model, recognized limitations, which raises concerns about its generalization. I believe that this issue should be better emphasized, in order to convey information more assertively.
2. The study omits a direct comparison with existing prognostic indicators for CESC. I understand that this comparison could be fundamental for some additional validation when making a comparison against other markers established in literature, addressing the effectiveness of the findings exposed here. I think this must be approached, so I have selected some of the references that may be interesting:
Xia WT, Qiu WR, Yu WK, Xu ZC, Zhang SH. Identifying TME signatures for cervical cancer prognosis based on GEO and TCGA databases. Heliyon. 2023 Apr 7;9(4):e15096. doi: 10.1016/j.heliyon.2023.e15096.
Xu T, Jiang J, Xiang X, Jahanshahi H, Zhang Y, Chen X, Li L. Conduction and validation of a novel prognostic signature in cervical cancer based on the necroptosis characteristic genes via integrating of multiomics data. Comput Biol Med. 2024 Jan;168:107656. doi: 10.1016/j.compbiomed.2023.107656.
Andalib KMS, Rahman MH, Habib A. Bioinformatics and cheminformatics approaches to identify pathways, molecular mechanisms and drug substances related to genetic basis of cervical cancer. J Biomol Struct Dyn. 2023;41(23):14232-14247. doi: 10.1080/07391102.2023.2179542.
Reza MS, Harun-Or-Roshid M, Islam MA, Hossen MA, Hossain MT, Feng S, Xi W, Mollah MNH, Wei Y. Bioinformatics Screening of Potential Biomarkers from mRNA Expression Profiles to Discover Drug Targets and Agents for Cervical Cancer. Int J Mol Sci. 2022 Apr 2;23(7):3968. doi: 10.3390/ijms23073968.3. Ethical considerations: particularly regarding patient consent, data privacy, and psychological impact, are not addressed and it seems to be that before implementing RNA-based prognostic models in clinical settings it should have been. Please clarify the ethical considerations of the study explicitly. I was not able to find it.
